# WAAT: a Workstation AR Authoring Tool for Industry 4.0

Category: Research

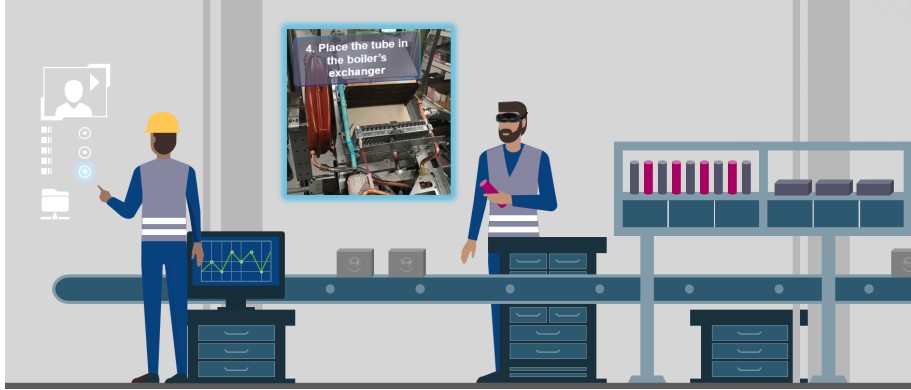

Figure 1: AR guidance in an assembly line

## ABSTRACT

The use of AR in an industrial context could help for the training of new operators. To be able to use an AR guidance system, we need a tool to quickly create a 3D representation of the assembly line and of its AR annotations. This tool should be very easy to use by an operator who is not an AR or VR specialist: typically the manager of the assembly line. This is why we proposed WAAT, a 3D authoring tool allowing users to quickly create 3D models of the workstations, and also test the AR guidance placement. WAAT makes on-site authoring possible, which should really help to have an accurate 3D representation of the assembly line. The verification of AR guidance should also be very useful to make sure everything is visible and doesn't interfere with technical tasks. In addition to these features, our future work will be directed in the deployment of WAAT into a real boiler assembly line to assess the usability of this solution.

**Index Terms:** Human-centered computing—Human computer interaction—Interaction paradigms—Mixed / augmented reality; Human-centered computing—Human computer interaction—Interaction paradigms—Virtual reality; Human-centered computing—Human computer interaction—Interaction design process and methods—Interface design prototyping;

## 1 INTRODUCTION

In a context of constant industrial evolution and a need of more industrial agility to answer the increasingly unpredictable customer requests, having well trained assembly line operators is extremely important. This operator training problematic is well known by the XXX company, a boiler manufacturer for which this training problem is very important during each winter, when boiler orders rise a lot, as many new operators must be hired for a few months.

After meetings and interviews with the XXX company, it appears that the training of these operators requires time and human resources. Currently, the training is done in 3 days with one experienced operator showing three trainees how to perform the technical gestures at their workstation. During these training days, none of the workers actually works on the assembly line and no boiler is built. After that, it takes 2 weeks for the operator in training to perform the technical tasks in the required time. Furthermore, only 1/3 of the new operators accept the job at the end of the training.

To improve the training of these new operators, we want to pro-

pose an AR-based operator training system, allowing the operators to train directly on the assembly line without the need of an experimented operator. This tool could also be used by experimented operator to train on a new position of the assembly line or on a different assembly line.

AR can be used in many different ways in the industrial context. It can be use to make expert remote guidance, create new products in collaborative engineering, or realize digital inspection of new prototypes [3]. AR is also used to test the ergonomics of the workstations and the reachability of some equipment. Testing new layout for the plant is also one of the AR use [1]. These different use are interesting, but the most interesting feature of AR and the most used is the assembly assistance, for training use or not [12, 15].

To use our AR-based operator training system we need to be able to create quickly and easily a 3D representation of the assembly line, and to be able to place anchors and markers (the details will be given in section 3.2.1) for the AR guidance elements. Furthermore, this tool must be very easy to use because the operators are not used to this technology. Indeed, the final users of the authoring tool will be the line managers, who are not used to use AR devices, or don't even know what AR is. That is why we need to adapt the interactions to the line manager, to ease the use of the tool.

The remaining of this paper is organized as follows: in section 2 we explain the industrial constraints for the use of this authoring tool. Section 3 contains the related work, justifying the choices made in section 4 which describes WAAT, our authoring tool. Finally, section 5 concludes this paper and discusses about future developments.

## 2 INDUSTRIAL CONTEXT

When working on an assembly line, every gesture is thought to accomplish the technical action in the smallest amount of time possible. Every workstation is thought to accomplish a certain number of technical tasks, the placement of each equipment is well thought. The authoring tool should help the operator gaining time or it won't be used in the factory. The time and space constraints have to be taken into account in the development of our tool.

### 2.1 Managing evolution

The XXX company assembly lines are manageable and can be changed during the day, rearranged to correspond with the boiler model being made. Each workstation can be rearranged (some parts

are on wheels to be easily movable). Some tasks can pass from a workstation to another when changing the boiler model being made.

This leads to the need of regular updates of the 3D model of the assembly line to correspond to the state of the real one and/or to new boilers models. The most suitable user to perform these updates is the line manager, who knows the assembly line state and each workstation very well.

## 2.2 Main features

Knowing these requirements, we have several uses for our authoring tool. First, the authoring tool must allow fast and easy placement of 3D models of the workstation (see Figure 2 (a)). The AR guidance panel and information must also be placed easily (see Figure 2 (b)). The AR part of the tool must allow the user to compare the 3D models and the real objects (see Figure 2 (c)), and to move or resize the 3D objects to match the real scene and the objects, and also make sure the operator guidance panel is visible (see Figure 2 (d)).

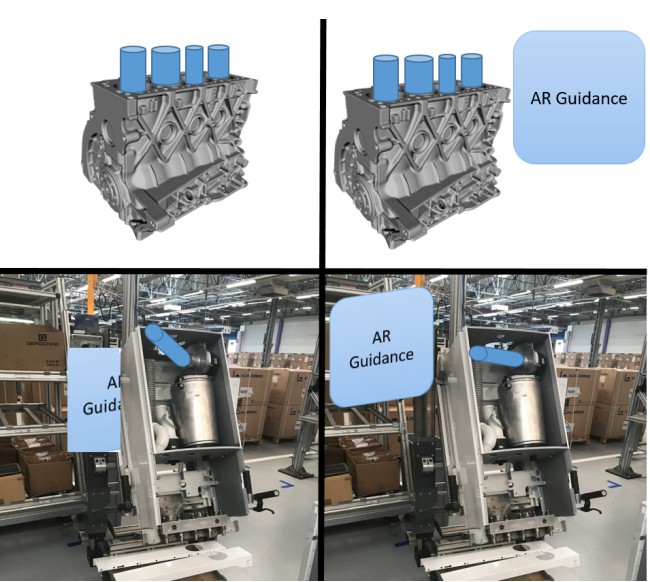

Figure 2: Needed features of the authoring tool. (a) 3D models placement with a motor. (b) AR guidance placement on the motor. (c) AR checking of 3D models and AR guidance on a real boiler. (d) AR validation of the placement on a real boiler

The next section presents the state of the art about 3D authoring tools that we could use to meet all our requirements.

## 3 RELATED WORK

## 3.1 3D Authoring tools

3D models of components and workstations are widely used in the industry. Most of the time they are used to create the full product, change the layout of the factory, test the ergonomics of the workstations [1]. To realize these 3D models, many tools are available since the 70s. Some of the most used tools to model industrial component are SolidWorks [1], Inventor [2], CATIA[3], Pro/ENGINEER [4] and AutoCAD [5]. These tools offer precise modeling features. The problem with these CAD (Computer Aided Design) tools is that their learning curve is very steep and require dozens of hours to properly use them [8]. It is also difficult to export files to AR engines like

Unreal [6] or Unity [7] without the use of an other software. Usually, the exported object are opened in a DCC (Digital Content Creation) tool (Blender [8] for example) to be converted in a suitable format for AR engines.

Other digital content creation tools, such as 3dsMax [9], Maya [10], Blender, can be used to model our workstation, but just as the previous CAD softwares seen previously, their interface is very complicated, and the learning time is very high. They offer lots of options to create very precise 3D models. SketchUp [11], another DCC tool, is easier to learn since it is based on line drawing but the 3D models obtained this way are less accurate than the others [13].

These 3D modeling tools are great when accurate 3D models are required, but they target professional designers who can spend time to learn a new tool and realize accurate 3D models of industrial components and workstations. In our case, we don't need such accurate 3D models. Indeed we only need to represent the components of the workstation and their position to the AR guidance tool.

## 3.2 AR Authoring

To be able to use AR authoring tools in our assembly lines, we have to know where to locate the 3D models in the space, to place them at the right position. To achieve that goal, several tracking techniques are used. These techniques are detailed in the next subsection.

### 3.2.1 Tracking methods

Tracking is a fundamental issue when using AR. Accurate and stable tracking techniques are important for using AR in an industrial environment. Most of the time, markers are used to locate the real objects and help positioning 3D objects. The capture of the marker is made by image recognition. After recognition, the 3D object linked with the marker appears [4, 6, 11, 14–17]. Markers can be pictures, QR code, colored paper, any kind of 2D unique-looking object. This technique is easy to implement and useful in an environment that can change depending on the assembly task such as our assembly line. An other tracking technique uses object recognition by comparing a 3D model of the object to the real object captured by the camera [2]. This technique requires precise 3D models of our objects and is useful in a static environment, which is not the case with our assembly line.

### 3.2.2 AR Authoring tools

With the emergence of AR, many tools have appeared to build AR applications. One of the first, ARToolkit [7], provides a marker-based registration using computer vision. ARToolKit and its variations all require the user to have C/C++ skills. More recently, Vuforia [12] a software development kit, provides computer vision technology to recognize and track planar images and 3D objects in real time. To use Vuforia, C# skills are necessary. This is the main problem with creating AR content. Most of the time low level AR libraries are used, requiring programming skills from the user.

The need for High level AR authoring tools for non-programmers is important. The earlier frameworks were often a plug-in of an other software, like DART [10], a plug-in for Adobe Director [13] (not supported anymore). DART was supporting the 3D editing part of the software with visual programming, such as dragging a 3D object in a scene and adding behavioral scripts to the object. Another GUI-based visual AR authoring tool, AMIRE [5] allows the user to describe interactions between objects with visual representations,

[1]https://www.solidworks.com/

[2]https://www.autodesk.com/products/inventor/overview

[3]https://www.3ds.com/products-services/catia/

[4]https://www.ptc.com/en/products/cad/pro-engineer

[5]https://www.autodesk.com/products/autocad/overview

[6]https://www.unrealengine.com/en-US/

[7]https://unity.com/

[8]https://www.blender.org/

[9]https://www.autodesk.com/products/3ds-max/overview

[10]https://www.autodesk.com/products/maya/overview

[11]https://www.sketchup.com/

[12]https://developer.vuforia.com/

[13]https://www.adobe.com/products/director.html

which can become complex. An other approach with a more rigid framework was APRIL [9] based on XML descriptions. APRIL is a plug-in of their own platform containing XML parser. Using XML can be quite hard for a user with no programming skills at all.

More recently, 3D GUI-based authoring tools have been studied. In ACARS [17], the authoring part is realized on desktop using a 3D model of the equipment. The maintenance technician can do on-site authoring using a tangible object to change the information's position. The authoring in SUGAR [4] is also realized on desktop. The maintenance expert can place AR information around the equipment using depth map of the factory. The authoring is also made by an expert in ARAUM [2], he gives information to the authoring tool about the procedures to perform, which will place automatically the AR content around the equipment using spatial, time and user context. The target devices for these authoring tools are HMD (Head Mounted Display) devices, but a mobile platform (tablet or phone) can also be a good alternative. Using the VEDILS framework [11] allows any user to create AR content for mobile using a desktop application. It uses visual programming and behavioral blocks to add interactions to the 3D objects, but it requires knowledge in algorithmic to be used properly.

Using mobiles to visualize AR content leads to research on mobile AR authoring tools, reducing the material needed for the authoring task, allowing in-situ authoring, which can be really interesting in some situation where we don't have access to a computer. HARATIO [14] allows the user to add 3D objects from a library in a scene, including behavior scripts. All actions are made with a radial menu. We can see limitations with mobile authoring tools like in [6] or [16]. In the first one, the user interacts with the 3D object on the touch screen, whereas in the second one the interactions are done directly with the markers which can be combined. With these tools we can only associate one marker with one 3D object.

It seems that authoring tasks made using desktop applications are often realized by an expert on the industrial procedure or a designer while we need our tool to be usable by line managers. It also seems that most authoring tools don't allow users to directly author the content while in AR to better match the real objects. Mobile authoring can be interesting in our case but the interaction limitations and the marker/3D object limit is a problem because we will have multiple 3D objects per workstation.

## 4 WAAT (WORKSTATION AR AUTHORING TOOL)

To meet all these requirements summarized in Table 1, we propose WAAT: an authoring tool designed to allow users to create 3D scenes of workstations by placing 3D objects representing the workstation. The originality of WAAT is that it makes it possible for the user to compare the 3D model with the real workstation.

### 4.1 System overview

WAAT is composed of two modules, a desktop 3D authoring module and an AR authoring module. The desktop authoring module is used to create the 3D scene whereas the AR authoring module is used to modify and validate the placement of the 3D models. The two modules are connected by the server, with a JSON file for each scene allowing the user to modify the scene either on desktop or AR.

WAAT is developed with Unity using Vuforia for the marker tracking in the AR module. The AR hardware currently used is the Microsoft Hololens [14]. To deploy WAAT on another AR device such as the Magic Leap [15] we would just have to add to our system the Magic Leap extension of the SDK (Software Development Kit) we currently use.

---

[14]https://www.microsoft.com/en-us/hololens
[15]https://www.magicleap.com/

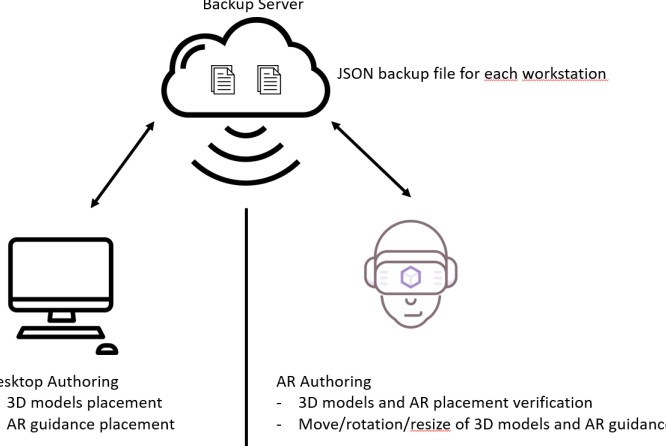

Figure 3: WAAT system overview

### 4.2 3D authoring of the assembly line

The authoring task on the desktop application is done with simple interactions using a mouse and a keyboard. To select which object the user wants to add, he just has to choose the object in the bottom interface (see Figure 4) and define its position in the scene. He can then move the objects to place them more precisely and change some of their properties (e.g: name, scale, . . . ). We chose drag and drop as the interaction method but it is not the only one available, and testing will help to choose the best suited one. Then the user can save the scene he created or load a scene previously created.

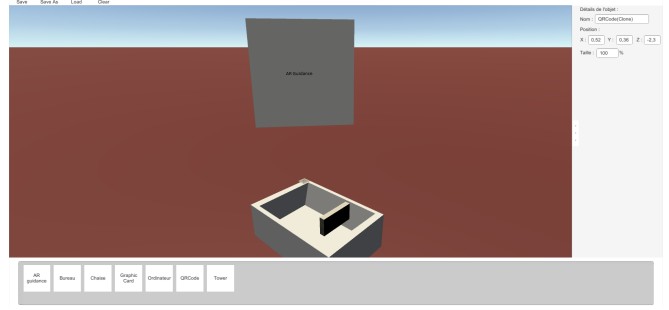

Figure 4: WAAT desktop module

For validation purpose without disturbing the workers and the whole assembly line, the task we tested the tool on was the installation of a graphic card in a computer (see Figure 5).

We conceived WAAT to also be usable in an immersive mode, but we focused on the main features of the tool and our users want to use it in non-immersive mode for the moment.

### 4.3 Using AR to check the 3D modeling

To make sure the 3D model of the assembly line is on par with the real assembly line, the user will use the AR part of the tool and compare the position, rotation, size of the 3D objects with their corresponding real object.

If it is not on par with the real objects (as illustrated in Figure 6), the user can move the 3D objects by grabbing them and placing them at the correct position. Each 3D objects refers to a real object and must be placed, rotated and scaled exactly as its real counterpart to allow the AR guidance system to be well configured. The AR

| Tool | 3D model placement | AR Guidance placement | AR authoring | Modular tracking | short learning time | GUI-based |
|------|--------------------|-----------------------|--------------|------------------|---------------------|-----------|
| SolidWorks | O | O | X | X | X | O |
| Inventor | O | O | X | X | X | O |
| CATIA | O | O | X | X | X | O |
| Pro/ENGINEER | O | O | X | X | X | O |
| AutoCAD | O | O | X | X | X | O |
| Blender | O | O | X | X | X | O |
| 3dsMax | O | O | X | X | X | O |
| Maya | O | O | X | X | X | O |
| SketchUp | O | O | X | X | X | O |
| ARToolKit | O | O | X | O | X | X |
| Vuforia | O | O | X | O | X | X |
| DART | O | O | X | O | - | O |
| AMIRE | O | O | X | O | - | O |
| APRIL | O | O | X | O | - | X |
| ACARS | O | O | O | O | - | O |
| SUGAR | O | O | X | O | - | O |
| ARAUM | O | O | X | X | - | O |
| VEDILS | O | O | X | O | X | O |
| HARATIO | O | O | O | X | O | O |
| Jung et al. | O | O | O | X | O | O |
| Yang et al. | O | O | O | X | O | O |

Table 1: Table summarizing the main features of 3D authoring tools suited for AR authoring.

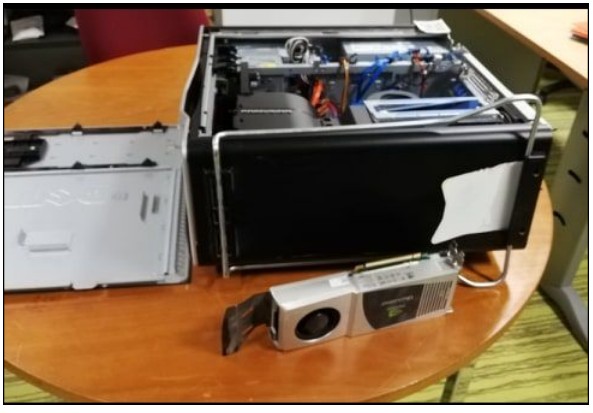

Figure 5: Technical task to realize, inserting the graphic card in the computer

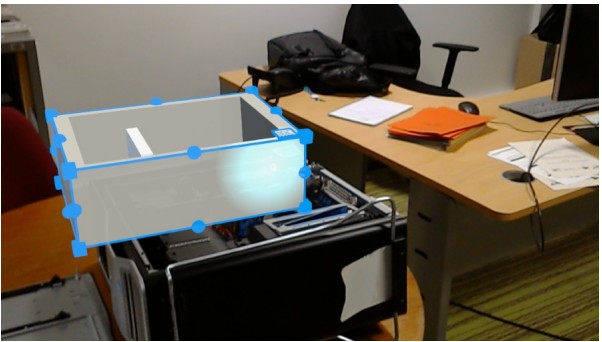

Figure 6: Bad placement of the 3D models

guidance system will use the position of the 3D objects to tell the operator where to grab or place technical parts.

After finishing all his modifications, the user can save his work, this will update the 3D model of the assembly line in the WAAT system. He can now load another scene to check the 3D modeling.

### 4.4 WAAT and AR guidance

WAAT is also used to place the marker position for further AR guidance of the operators. Each scene is represented by a marker allowing the system to load the correct objects related to a workstation. The 3D objects in WAAT are used as anchors for the AR guidance system to help the operator in training.

#### 4.4.1 Placing AR markers in the model of the assembly line

After the 3D modeling of the assembly line, the final step is to place the markers positions in the virtual scenes. This virtual marker is used to show the user in AR the position where he is supposed to place the real marker to use the system. It is placed in the same way than any other 3D object.

#### 4.4.2 Placing AR markers in the real assembly line

To be able to test the AR guidance (and to use it during the operator formation) the user has to place real markers at the same place as the "virtual" markers in the real assembly line. For every workstation, a specific marker is placed in a specific position. If the real marker is not at the same position as the virtual one, all the 3D objects will be displaced compared to the real objects. If the position of the virtual marker is too difficult to use in the assembly line, the user can move it to match the correct position.

#### 4.4.3 AR Checking of the AR guidance

The AR scene is also used to verify the positioning of the "operator guidance": the panels where the training instructions will be shown. The user can move these panels to make sure the instructions are visible and that there is no occlusion or other problems. The user will also check the positioning of all markers and objects and move them if needed since they will also be used by the AR guidance system.

Here again, after finishing all his modifications, the user can save his work, this will update the assembly line model globally, and more precisely, the markers positions and AR guidance panels positions.

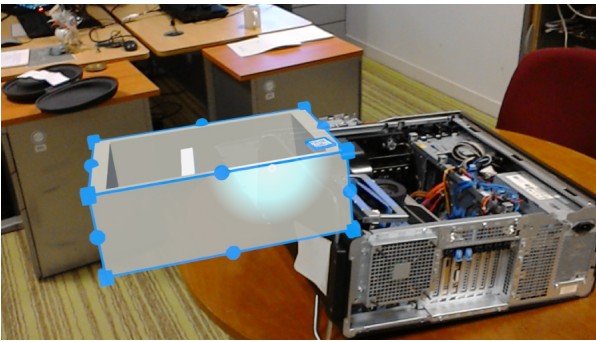

Figure 7: Bad marker placement on the real object

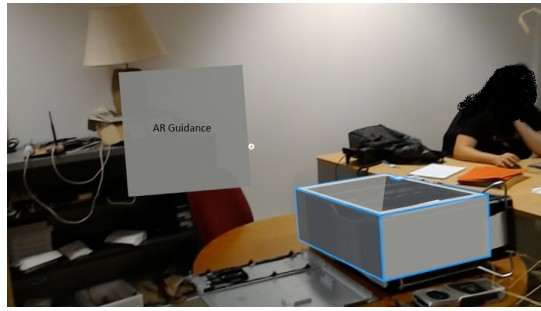

Figure 8: Good placement of the 3D models and the AR Guidance

## 5  CONCLUSION

So we created WAAT, a 3D authoring tool allowing untrained users to create 3D models of the assembly line in a boiler factory. WAAT allow fast creation and AR comparison of the 3D and the real model. In case of a mismatch, the user can move the 3D components to match the real workstation. He can test the AR guidance used to train assembly line operators to make sure everything is fully visible and doesn't interfere in the realization of the technical tasks. Every modification made in desktop and AR is saved in a file on a server so there is no need to change on multiple platform.

WAAT needs to be tested with 3D elements corresponding to the factory's workstations to find the most suitable level of details the 3D objects need to have, and the level of fidelity in the positioning of the 3D objects.

We also have to test this tool with real operators from the factory to test the usability and the intuitiveness of WAAT.

To adapt the interactions used to move/rotate/resize the 3D objects to our users, adding accessories such as a controller will be explored. We will also test new AR systems to test their usability and native interactions to see if we can use them, or if we can adapt them to our users.

The immersive mode of WAAT will allow us to visualize the augmented environment and test the placement of the operator guidance, even if it's not the focus right now. We will be able to simulate the AR test of the scenes without the need to be in the plant all the time.

## ACKNOWLEDGEMENTS

This work is possible thanks to the workers of the XXX.YYY plant and of XXX.ZZZ plant for exchanging with us on the problematic of their work and helping us to better understand their needs in 3D modeling and AR guidance.

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
