# OpenReview forum: "WAAT: a Workstation AR Authoring Tool for Industry 4.0"
_graphicsinterface.org/Graphics_Interface/2020/Conference — Submitted to GI 2020_

### Official Review · AnonReviewer2 · 2020-01-04
**Lacking clear research contribution and framing.**

**Confidence:** 5
**Rating:** 2

**Review:**

This paper presents a system that allows users to place and position 3D objects in augmented reality. The system is motivated by a need to train factory workers at an increased rate, and demonstrated through a system which helps users align a graphics card into a computer.

This paper, while it addresses an important area (improving training for skilled workers), doesn't appear to make a clear research contribution.

Using augmented reality for training workers is a widely studied area that has been examined for over a decade.  Boud, Andrew C., et al. "Virtual reality and augmented reality as a training tool for assembly tasks." 1999 IEEE International Conference on Information Visualization (Cat. No. PR00210). IEEE, 1999.

This paper does not contrast itself to the vast array of prior work, or make any distinguishable contribution. The idea of using AR to guide workers is very well explored.

There is also no evaluation, simply a presentation of ideas and screenshots of an early prototype. The paper reads more like a technical report for a company than an academic research paper. Much more work needs to go into this before it is ready for publication.

---

### Official Review · AnonReviewer3 · 2020-01-06
**Work-in-progress**

**Confidence:** 4
**Rating:** 3

**Review:**

The paper "WAAT: a Workstation AR Authoring Tool for Industry 4.0" presents an AR guidance system for industrial assembly lines that allows for on-site authoring of AR content. The system allows for the placement of 3D models, guidance widgets, and calibration of placement in relation to physical counterparts. The system is not evaluated in-situ, but a small validation was done in the lab where the task was installing a graphics card in a computer.

The topic addressed by the paper is important and timely as the technology for augmented reality is maturing and the big commercial players such as Apple, Microsoft and Google are starting to pick it up. However, augmented reality for assembly tasks has been a core vision in AR research for over two decades and manufacturing is one of the domains highlighted on the Vuforia website (Vuforia is the software development kit for AR used by the authors). Therefore, I would have liked to see the authors present the current state of the art in AR for assembly line tasks and how their work relates. A quick Google Scholar search on "augmented reality assembly line" reveals dozens of papers some of which dating back to the early nineties.

The paper provides no real evaluation of the system and technique. The authors have made a brief preliminary evaluation in the lab on an artificial task. The authors state in the conclusion that for future work the system needs to be tested with elements corresponding to the workstation at the factory and it should be tested with real operators. I agree with this sentiment, but without this evaluation, the paper remains a work-in-progress paper.

Therefore my conclusion is that the work is in a too preliminary state for publication and that the authors will need to position their work better towards previous research on AR for assembly lines.

---

### Official Review · AnonReviewer1 · 2020-01-07
**The paper presents an AR authoring system for assembly line systems**

**Confidence:** 4
**Rating:** 3

**Review:**

The paper addresses an important area - how can AR authoring tools support training of assembly line systems? The approach taken by the paper is promising. However, this seems like a work in progress.

The paper describes a very generic system and it is unclear how this design separates it from other works. Why can't one design the 3D models in Solidworks and then place those 3D models in AR using another AR tool. Why does everything need to be done with a single tool? The 3D modelling tool described seems nowhere close to professional 3D modelling tools.

How does the system address the specific challenges of the assemble line? What interaction flows can it support?

The paper presents no evaluations of the system as well.

---

### Meta-Review · Area_Chair1 · 2020-01-10

**Recommendation:** Reject
**Confidence:** 5

**Metareview:**

The reviewers agree that the paper addresses an important research area, but they also agree that what the paper reports on is work-in-progress and lacks evaluation and a proper positioning in relation to previous work on AR for assembly tasks.

The reviewers unanimously agree that the paper should be rejected.

---

### Decision · Program_Chairs · 2020-01-11

Reject